behaviour, cognition, ecology

gesture, vocalization, universal, icon, language origin, language evolution

**Author for correspondence:**
Nicolas Fay
e-mail: nicolas.fay@gmail.com

# Gesture is the primary modality for language creation

Nicolas Fay[1], Bradley Walker[1], T. Mark Ellison[2], Zachary Blundell[1], Naomi De Kleine[1], Murray Garde[3], Casey J. Lister[1] and Susan Goldin-Meadow[4]

[1]School of Psychological Science, University of Western Australia, 35 Stirling Highway, Crawley, WA 6009, Australia
[2]Collaborative Research Centre for Linguistic Prominence, University of Cologne, Cologne, NRW, Germany
[3]School of Culture, History and Language, College of Asia and the Pacific, Australian National University, Canberra, ACT, Australia
[4]Department of Psychology, University of Chicago, Chicago, IL, USA

NF, 0000-0001-9866-2800

How language began is one of the oldest questions in science, but theories remain speculative due to a lack of direct evidence. Here, we report two experiments that generate empirical evidence to inform gesture-first and vocal-first theories of language origin; in each, we tested modern humans' ability to communicate a range of meanings (995 distinct words) using either gesture or non-linguistic vocalization. Experiment 1 is a cross-cultural study, with signal Producers sampled from Australia ($n = 30$, $M_{age} = 32.63$, s.d. = 12.42) and Vanuatu ($n = 30$, $M_{age} = 32.40$, s.d. = 11.76). Experiment 2 is a cross-experiential study in which Producers were either sighted ($n = 10$, $M_{age} = 39.60$, s.d. = 11.18) or severely vision-impaired ($n = 10$, $M_{age} = 39.40$, s.d. = 10.37). A group of undergraduate student Interpreters guessed the meaning of the signals created by the Producers ($n = 140$). Communication success was substantially higher in the gesture modality than the vocal modality (twice as high overall; 61.17% versus 29.04% success). This was true within cultures, across cultures and even for the signals produced by severely vision-impaired participants. The success of gesture is attributed in part to its greater universality (i.e. similarity in form across different Producers). Our results support the hypothesis that gesture is the primary modality for language creation.

## 1. Introduction

People of all cultures gesture while they speak [1,2], blind people gesture [3], and hearing adults and children can successfully use gesture as their sole means of communication at the request of experimenters [4–8]. Furthermore, sophisticated manual languages, with the same expressive range as spoken language [9], emerge rapidly in populations of deaf children [10,11]—and even among individual deaf children living in hearing households [12]—or in communities with a high incidence of deafness [13]. The ubiquity of gesture, and its capacity to rapidly evolve into language, has led to the proposal that language originated in manual gestures rather than in vocal calls [14,15]. The present study tests this proposal with two experiments.

The gesture-first theory of language origin dates back to the eighteenth century [14,15], but has recently gained in popularity [16–20]. Consistent with a gesture-first theory, comparative studies have demonstrated greater flexibility in non-human primates' (hereafter: primates) gestures compared to vocal calls [21], more success in teaching primates sign language than vocal language [22–24], and striking similarities between the naturalistic gestures produced by young children and by chimpanzees [25]. Support for the vocal-first theory of language origin [26–28] includes comparative evidence indicating that primates use vocal calls to convey specific information to conspecifics [29], primate vocal

calls are more flexible than first thought by gesture-first proponents [30], and primates can expand their limited vocal repertoire by combining single calls into structurally more complex units with a different meaning (a possible precursor to syntax; [31,32]). Comparative studies therefore offer support for gesture-first and vocal-first theories of language origin. However, because meaning is operationalized differently in primate vocal and gesture studies, the findings cannot currently be compared across the modalities [33].

Laboratory experiments are increasingly being used to test the factors that drive language evolution (e.g. [34–36]) (with the caveat that they rely on modern human participants). To inform gesture- and vocal-first theories of language origin, researchers have compared modern humans' ability to communicate a range of experimenter-specified meanings when communication is restricted to the gesture modality or the non-linguistic vocal modality (hereafter: vocal modality). The basic finding is that while vocalization can successfully convey the different meanings at above chance levels [37–40], communication success is considerably higher in the gesture modality [41–43]. This pattern of results supports a gesture-first theory of language origin. However, the generalizability of these findings is limited by their reliance on culturally homogeneous samples of participants, namely those from Western, educated, industrialized, rich and democratic (WEIRD) societies [44], and by the small, potentially unrepresentative sample of meanings participants communicated (between 4 [45] and 27 [36] distinct meanings). The experiments reported here aim to overcome these issues.

A common explanation for the communication success of gesture is its affordance for iconic signal production (i.e. signals that perceptually resemble their referent) [41,46–48]. Gesture and vocal studies indicate that communication success is positively correlated with signal iconicity, and because signal iconicity is higher for gestured signals than for vocal signals, communication success is higher in the gesture modality [37,38,49]. However, a recent study indicates that the greater universality of gestured signals (i.e. the degree to which different people produce similar signals to convey the same meaning) may also contribute to their communication success [50]. In this study, children's communication success was positively correlated with the extent to which their spontaneous signals (gesture or vocal) resembled the signals produced by adults for the same meaning, and because signal universality was higher for gestured signals than for vocal signals communication success was higher in the gesture modality. Importantly, although signal universality was moderately correlated with signal iconicity, it independently predicted communication success. The universality of gestured signals has long been recognized by philosophers and early explorers, who recommended using 'the universal language of the hands' to communicate with indigenous people (Quintilian, 95 CE [51]). The basic idea is that people of different cultures can successfully communicate because they represent meaning similarly in the gesture modality. The universality of gesture and vocal signals, and their relationship to communication success, is tested in the experiments reported.

In the experiments reported, participants (Producers) communicated a variety of experimenter-specified meanings using either gesture or vocalization. The signals they created were recorded and replayed to a second group of participants (Interpreters) who tried to guess the meaning of each signal. This provided our measure of communication success. Across

participants, a large number of meanings were sampled (180 words in Experiment 1 and 815 words in Experiment 2) to ensure the results generalized across meanings. Experiment 1 is a *cross-cultural* study that sampled Producers from two very different cultures, a WEIRD culture (Australia) and a non-WEIRD subsistence-agriculturalist culture (Vanuatu). If gesture is the primary modality for language creation, then communication success will be higher for gesture than for vocalization, both within cultures and across cultures. If gesture is a universal means of communication—as predicted by philosophers and early explorers—then the gestures produced by different participants will be more similar in form than the corresponding vocalizations, both within and across cultures. If signal universality aids communication success then the two will be positively correlated.

Experiment 2 tests *why* gesture may be a more universal means of communication than vocalization (i.e. the mechanism underlying signal universality). This is predicted by an embodied account of cognition, which highlights the importance of the body, and the body's interactions with the environment, to cognition [52–55]. To the extent that the experimental Producers share the same body plan and use their body to physically interact with their environment in similar ways (e.g. drinking by raising a container to their mouth), they should produce gestures that resemble one another (e.g. communicating 'drink' via manual simulation). By contrast, the opportunity for signal embodiment is absent in the vocal modality. Experiment 2 is a *cross-experiential* study. It is identical in design to Experiment 1, but the Producers were either sighted or severely vision-impaired. Vision-impaired Producers cannot rely on their visual experience (e.g. socially learned gestures) and must instead rely on their physical interactions in the environment. If embodiment is important to signal universality then, like sighted Producers, vision-impaired Producers will show greater evidence for signal universality in the gesture modality, compared to the vocal modality (despite their shared auditory experience and the absence of a shared visual experience). If the effect of embodiment on signal universality can explain the differential communication success of the gesture and vocal modalities, signal universality will again be positively correlated with communication success.

## 2. Methods

### (a) Experiment 1: cross-cultural

#### (i) Participants

Thirty participants from Vanuatu (27 males and 3 females, $M_{age}$ = 32.40, s.d. = 11.76, 18–51 years) and 30 participants from Australia (matched on age and gender; 27 males and 3 females, $M_{age}$ = 32.63, s.d. = 12.42, 18–53 years) were recruited as Producers. An Australian participant was considered a match to a Ni-Vanuatu participant if they were within 5 years on age and were of the same gender. The Ni-Vanuatu participants were mostly from Pentecost Island, several of whom were members of a band that was visiting Port Vila (Vanuatu) to record a music CD. Members of this group are subsistence agriculturalists (supplemented with cash crops such as kava and toro), who have had up to 6 years of formal education and minimal contact with Western culture. Each participant was paid the equivalent of AU$10.

A convenience sample of 50 Australian undergraduate students from the University of Western Australia (17 males and

royalsocietypublishing.org/journal/rspb　Proc. R. Soc. B **289**: 20220066

33 females, $M_{age}$ = 19.86, s.d. = 3.08, 18–32 years) were recruited as Interpreters and tried to guess the words communicated by the Producers. Each Interpreter participated in exchange for partial course credit.

### (ii) Materials

Pilot testing indicated that lower-frequency words (e.g. 'republic') were often unfamiliar to the Ni-Vanuatu participants. We therefore created a corpus of 180 Basic English words (60 nouns, 60 verbs and 60 adjectives, based on Ogden [56]; see electronic supplementary material). From this corpus, a list of 36 words (12 nouns, 12 verbs and 12 adjectives) were sampled without replacement for each Producer. Each word was sampled multiple times as a target word in each modality.

### (iii) Task and procedure

#### Production phase

The referential communication task is similar to that used by Fay *et al.* [41,42]. Each Producer was seated in front of a video camera. The experimenter read out each to-be-communicated word and indicated its class (noun, verb and adjective) and the modality (gesture and vocal) to be used to communicate the word. All communication with the Ni-Vanuatu participants was in Bislama (author M.G. acted as translator). Author M.G. translated the English instructions and target words into Bislama for the Ni-Vanuatu participants. Producers could skip the trial if they felt unable to communicate the word in the specified modality. This was rare ($M_{skipped}$ = 1.85%, s.d. = 13.48%).

The production phase was administered over 2 blocks, with 18 words in each block. Producers were allowed to rest between blocks. The communication modality (gesture and vocal) alternated over blocks (with modality randomly assigned at Block 1). In the gesture condition, Producers were told that all communication was limited to gesture (hand, body and face) and vocalizing was prohibited. In the vocalization condition, Producers were told they could make any sounds they wished, but they were not permitted to use words. Across blocks, each Producer communicated 18 words using gesture and 18 words using vocalization, with an equal number of nouns, verbs and adjectives in each modality.

#### Interpretation phase

Each Interpreter was presented with a subset of recordings from the production phase of Experiment 1 on a computer (approximately half gesture, half vocal). Interpreters guessed the word being communicated from a list of 16 targets. Gesture recordings included no sound, and vocal recordings included no video. Interpreters could replay the recording as often as they wished. The recordings were sampled such that each Interpreter viewed a maximum of four recordings from a single Producer (a maximum of 2 from each modality). Across Interpreters each recording (2120 in total) appeared five times. The number of signals guessed by each Interpreter ranged between 204 and 216. After the Interpreter had guessed the word being communicated, they rated how confident they were that they had guessed correctly. This was done on a scale of 0 (*not at all confident*) to 6 (*extremely confident*). In total, Interpreters made 10 600 assessments.

### (iv) Measures

#### Skipped trials

Producers could skip a trial if they felt unable to communicate the word in the specified modality. Allowing participants to skip trials served two purposes: (i) it indexed how difficult participants found communication in each modality, and (ii) in the context of the cross-cultural study (Experiment 1), it mitigated any bias in the corpus of words (e.g. Ni-Vanuatu Producers

could skip words they were unfamiliar with). Skipped trials were coded as 1 and non-skipped trials were coded as 0.

#### Communication success

Communication was successful when the Producer's intended word was selected by the Interpreter (always from Australia) from a list of alternatives. The decision to use a multiple-choice format meant that Interpreters' guessing accuracy could be objectively assessed. If our design had allowed Interpreters to make open-ended guesses, correct guesses would be more difficult to objectively code as some guesses may be similar (but not identical) to the target word. We note that communication success may be inflated in the multiple-choice format used, relative to a more open-ended format (although the inflation should be the same for both the gesture and vocalization trials). Successful communication was coded as 1 and unsuccessful communication was coded as 0. Skipped trials were not included.

#### Interpreter confidence

Interpreter confidence was rated on a Likert scale from 0 (*not at all confident*) to 6 (*extremely confident*).

#### Signal universality

Signal universality was operationalized as the extent to which Producers used similar signals to communicate the same words. This was measured by rating the similarity of the signals produced for the same word by different Producers (done separately for each modality). For each signal, we randomly selected a signal produced by another Producer from the same culture for the same word, and a signal produced by a Producer from the other culture for that word, and compared it to each of these. Pairs of signals were rated on a Likert scale from 0 (*extremely dissimilar*) to 6 (*extremely similar*) (5574 pairings) by author C.J.L. A second coder (author B.W.) rated a random selection of 20% of the pairs of signals using the same procedure (1115 pairings). Inter-coder reliability indicated substantial agreement (Krippendorff's alpha = 0.714) [57]. A baseline measure of signal similarity was computed by using the same procedure but for signals produced for different words (5574 pairings). Three dyad combinations were created: Australia–Australia, Vanuatu–Vanuatu and Australia–Vanuatu.

### (v) Statistical analysis

The data were analysed using logistic and cumulative link mixed effects modelling. All analyses were performed and all figures were created in R [58]. Statistical models were estimated using the glmer() function of lme4 [59] (for binary data: skipped trials and communication success) and the clmm() function of the ordinal [60] package (for Likert data: Interpreter confidence and signal universality). For each analysis, the maximal random effects structure justified by the experiment design was specified where possible [61]. Descriptive statistics and the output from the statistical models, including the specification of the random effects structure, are provided in the electronic supplementary materials. The data and R scripts associated with Experiment 1 and 2 are available on the Open Science Framework (OSF): https://osf.io/36jpy/.

## (b) Experiment 2: cross-experiential

### (i) Participants

Ten severely vision-impaired participants (6 males and 4 females, $M_{age}$ = 39.40, s.d. = 10.37, 22–53 years) and 10 sighted participants (matched on age, gender and education level; 6 males and 4 females, $M_{age}$ = 39.60, s.d. = 11.18, 22–54 years) were recruited as Producers (all Australian). Each was paid AU$10. All participants were native English speakers. The smaller

sample size in Experiment 2 compared to Experiment 1 was due to the difficulty in recruiting severely vision-impaired participants. The average age at which vision was lost for the vision-impaired participants was 1.8 years old (s.d. = 3.6). Six were blind (5 from birth) and all had at most 10% visual acuity or light perception at the time of the study. This group is referred to as the 'Vision-Impaired' group. A sighted participant was considered a match to a vision-impaired participant if they were within 5 years on age, were of the same gender and had the same level of education (no tertiary education, undertaking tertiary education or completed tertiary education). This group is referred to as the 'Sighted' group.

A convenience sample of 90 Australian undergraduate students from the University of Western Australia (22 males and 68 females, $M_{age}$ = 21.11, s.d. = 6.57, 17–54 years) tried to guess the words being communicated by the Producers. Each Interpreter participated in exchange for partial course credit.

### (ii) Materials
A corpus of 1260 words (420 nouns, 420 verbs and 420 adjectives) was sampled from the 5000 most commonly used words in American English, based on the Corpus of Contemporary American English [62] (see electronic supplementary materials). From this corpus, a list of 72 words (24 nouns, 24 verbs and 24 adjectives) were sampled without replacement for each Producer. In total, 815 unique words were sampled as target words. Sampling Producers from the same culture allowed us to use a much larger corpus of words compared to Experiment 1 (where we sampled a smaller corpus of basic words to ensure their meanings were shared across cultures).

### (iii) Task and procedure
#### Production phase
The production phase is similar to Experiment 1. With a more complex word list in Experiment 2, Producers skipped a higher proportion of trials compared to Experiment 1 ($M_{skipped}$ = 22.15%, s.d. = 41.54%). This reduced the number of unique words communicated by Producers to 649. Because of the smaller sample of Producers in Experiment 2, we asked each Producer to communicate more words (72 words per Producer). The production phase was administered over four blocks, with 18 words in each block. As before, modality alternated across blocks (with modality randomly assigned at Block 1) and Producers were allowed to rest between blocks. Across blocks, each Producer was asked to communicate 36 words using gesture and 36 words using vocalization, with an equal number of nouns, verbs and adjectives in each modality.

#### Interpretation phase
The interpretation phase for Experiment 2 matched the interpretation phase for Experiment 1. Across Interpreters, each recording (1080 in total) appeared five times.[1] The recordings were sampled such that each Interpreter viewed a maximum of four recordings from a single Producer (a maximum of two from each modality), and each recording communicated a different word. The number of recordings viewed by each Interpreter ranged between 53 and 64. In total, Interpreters made 5399 assessments.

### (iv) Measures and statistical analysis
The same measures (skipped trials, communication success, Interpreter confidence and signal universality) and statistical analyses used in Experiment 1 were used in Experiment 2.

#### Signal universality
Because of the large number of words sampled, each word recurred less often in Experiment 2 than in Experiment 1. Signal similarity

ratings were therefore made for every pair of same-word signals produced in each modality (732 pairings). A baseline measure of signal similarity was computed by pairing each signal in the original set of comparisons with a signal for a different word in the same modality (1468 pairings). Pairs of signals were again rated on a Likert scale from 0 (*extremely dissimilar*) to 6 (*extremely similar*) by author CL. Three dyad combinations were created: sighted–sighted, vision-impaired–vision-impaired and sighted–vision-impaired.

## 3. Results
### (a) Experiment 1: cross-cultural results
#### (i) Skipped trials, communication success and interpreter confidence
Trial skipping in Experiment 1 was rare (1.85% of trials) so was not analysed (figure 1a). For communication success, the best-fitting model included fixed effects for Culture ($B$ = 0.91, s.e. = 0.09, $z$ = 10.36, $p < 0.001$), Modality ($B$ = 1.86, s.e. = 0.08, $z$ = 23.39, $p < 0.001$) and their interaction ($B$ = 0.43, s.e. = 0.16, $z$ = 2.75, $p = 0.006$). The effect of Culture demonstrates that the Australian Producers' signals were more accurately interpreted (by Australian Interpreters) than the Ni-Vanuatu Producers' signals. The effect of Modality demonstrates that, for Producers from both cultures, communication was more successful in the gesture modality than in the vocal modality. The Culture by Modality interaction reflects the greater benefit of gesture over vocalization when communication occurred across cultures ($B$ = 2.09, s.e. = 0.13, $z$ = 15.58, $p < 0.001$), compared to within cultures ($B$ = 1.64, s.e. = 0.09, $z$ = 17.58, $p < 0.001$) (figure 1b). The Interpreter confidence results mirrored the communication success results (figure 1c).

#### (ii) Signal universality
The best-fitting model included fixed effects for Culture ($ps < 0.001$), Modality ($B$ = 1.08, s.e. = 0.12, $t$ = 9.32, $p < 0.001$) and their interaction ($ps < 0.001$). As predicted, for each dyad combination (Australia–Australia, Vanuatu–Vanuatu and Australia–Vanuatu), pairs of gestured signals were more similar than pairs of vocal signals ($ps < 0.001$). The gestured signals were more similar in the Australia–Australia group compared to the Vanuatu–Vanuatu group ($p = 0.013$), which were more similar than the Australia–Vanuatu group ($p < 0.001$). The vocal signals were more similar in the Australia–Australia group compared to the Vanuatu–Vanuatu group ($p < 0.001$), but there was no evidence of a statistical difference between the Vanuatu–Vanuatu group and the Australia–Vanuatu group ($p = 0.237$). This difference in pattern explains the Culture by Modality interaction (figure 1d).

If the greater communication success of gesture is driven by its greater signal universality, signal universality will be positively correlated with communication success (i.e. a one-tailed test). When the data are combined across the within-culture groups (Australia–Australia and Vanuatu–Vanuatu) there are positive correlations between signal universality and communication success for gesture ($r_{58}$ = 0.49, $p < 0.001$) and for vocalization ($r_{58}$ = 0.66, $p < 0.001$). A similar pattern is seen in the across-culture Australia–Vanuatu group: gesture ($r_{58}$ = 0.36, $p = 0.002$) and vocalization ($r_{58}$ = 0.40, $p < 0.001$; figure 1e).

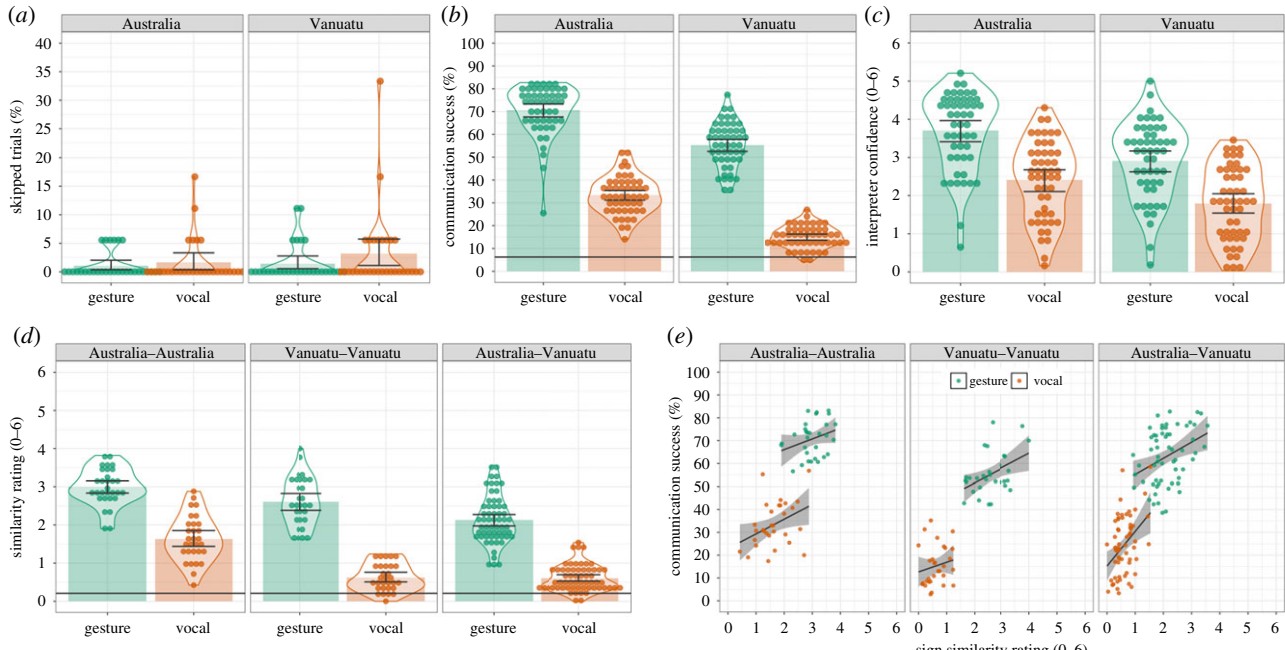

**Figure 1.** Summary statistics and distributional information for all dependent variables. (*a*–*d*) Bars indicate the cell means and the data points indicate the mean score for each Producer/Interpreter, error bars are the 95% bootstrapped confidence intervals and violins provide distributional information. (*a*) Percentage of words Producers skipped. (*b*) Percentage of words successfully communicated (organized by Interpreters; the black horizontal line indicates chance communication success). (*c*) Interpreters' confidence they had identified the Producers' intended word. (*d*) (Rated) similarity of the signals used to communicate the same words by different Producers (the black horizontal line indicates the baseline similarity). (*e*) shows the correlation between signal universality and communication success (the data points indicate the mean scores for each Producer). The black straight line is the linear model fit, and the grey shaded area is the bootstrapped 95% CI. Note that, for each condition, communication success was higher than chance (*b*) and signal similarity was higher than baseline signal similarity (*d*). (Online version in colour.)

## (b) Experiment 2: cross-experiential results

### (i) Skipped trials, communication success and interpreter confidence

For skipped trials, the best-fitting model included Modality as a fixed effect ($B = 1.05$, s.e. $= 0.37$, $z = 2.84$, $p = 0.004$). Producers skipped more trials in the vocal modality than in the gesture modality, indicating that they found communication in the vocal modality more difficult (figure 2*a*).[2]

For communication success, the best-fitting model included fixed effects for Group ($B = 0.59$, s.e. $= 0.20$, $z = 2.91$, $p = 0.004$), Modality ($B = 1.12$, s.e. $= 0.15$, $z = 7.31$, $p < 0.001$) and their interaction ($B = 1.02$, s.e. $= 0.30$, $z = 3.36$, $p < 0.001$). The effect of Group demonstrates that communication success was higher for sighted Producers than for vision-impaired Producers. The effect of Modality demonstrates that communication was more successful in the gesture modality than the vocal modality (despite Producers skipping more trials in the vocal modality). This was true for sighted and vision-impaired Producers ($ps < 0.030$). The Group by Modality interaction reflects the fact that communication success was greater for sighted Producers than vision-impaired Producers in the gesture modality ($B = 1.04$, s.e. $= 0.24$, $z = 4.40$, $p < 0.001$), but not in the vocal modality ($p = 0.928$) (figure 2*b*). The Interpreter confidence results mirrored the communication success results (figure 2*c*).

### (ii) Signal universality

The best-fitting model specified Modality as a fixed effect ($B = 0.76$, s.e. $= 0.17$, $z = 4.37$, $p < 0.001$). As predicted, signal similarity was higher for gesture than for vocalization for each dyad combination (figure 2*d*).

If gesture's success is driven by its greater universality, signal universality will be positively correlated with communication success (i.e. a one-tailed test). When the data are combined across the within-experience groups (sighted–sighted and vision-impaired–vision-impaired), there are positive correlations between signal universality and communication success for gesture ($r_{18} = 0.25$, $p = 0.148$) and for vocalization ($r_{18} = 0.66$, $p < 0.001$). A similar pattern is seen in the across-experience sighted—vision-impaired group: gesture ($r_{18} = 0.55$, $p = 0.006$) and vocalization ($r_{17} = 0.65$, $p = 0.001$; figure 2*e*).

## 4. Discussion

A primary function of language is to communicate meanings across people [63–65], and the experiments reported show that when people are unable to use language to communicate, gesture is better adapted to meaning transfer than (non-linguistic) vocalization. This was true within cultures and across cultures (Experiment 1), and even for the gestured signals produced by severely vision-impaired participants (Experiment 2). These findings are consistent with a gesture-first theory of language origin.

The greater affordance of the manual-visual modality for iconic signal production is a common explanation for gesture's greater success at bootstrapping human communication than vocalization [41,46–48]. Our results point to a second explanation: gesture is more successful than vocalization because gestured signals are more *universal* than vocal signals. Producers' gestured signals were more similar to each other than their vocal signals, both within cultures

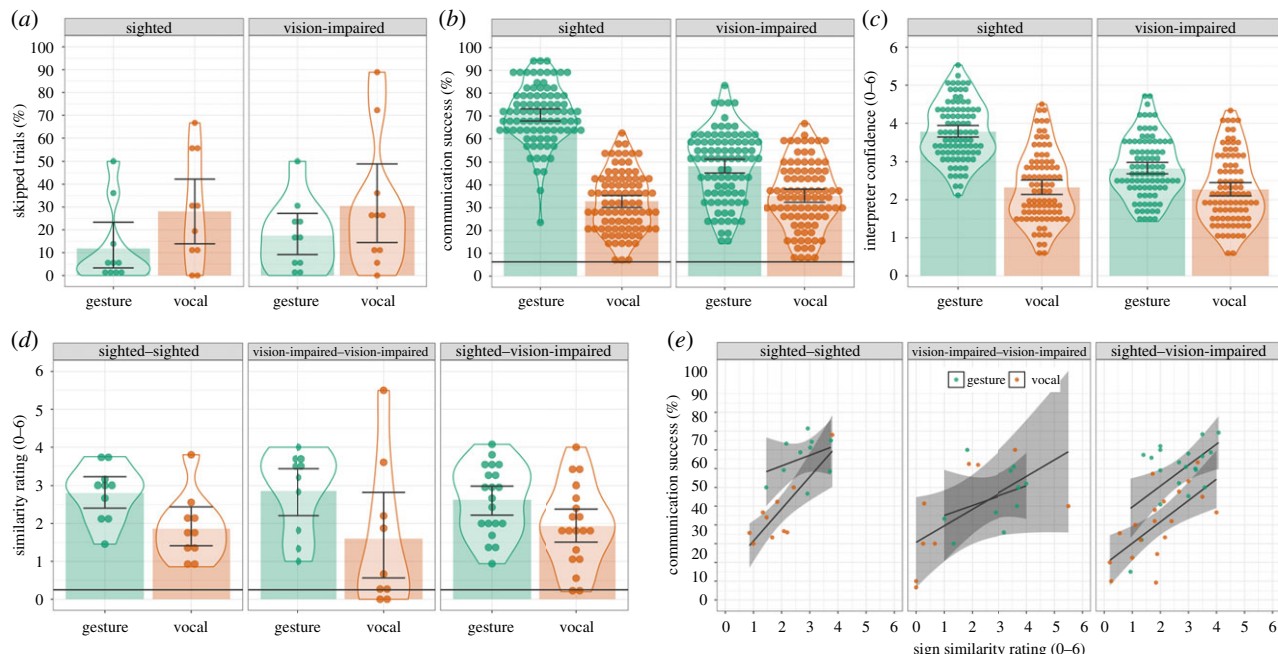

**Figure 2.** Summary statistics and distributional information for all dependent variables. (*a–d*) Bars indicate the cell means and the data points indicate the mean score for each Producer/Interpreter, error bars are the 95% bootstrapped confidence intervals and violins provide distributional information. (*a*) Percentage of words Producers skipped. (*b*) Percentage of words successfully communicated (organized by Interpreters; the black horizontal line indicates chance communication success). (*c*) Interpreters' confidence they had identified the Producers' intended word. (*d*) (Rated) similarity of the signals used to communicate the same words by different Producers (the black horizontal line indicates the baseline similarity). (*e*) Correlation between signal universality and communication success (the data points indicate the mean scores for each Producer). The black straight line is the linear model fit, and the grey shaded area is the bootstrapped 95% CI. Note that for each condition, communication success was higher than chance (*b*) and signal similarity was higher than baseline signal similarity (*d*). (Online version in colour.)

and across cultures. Strikingly, the gestured signals produced by severely vision-impaired participants were more similar to those produced by sighted participants than their vocal signals, despite their shared auditory experience, and the absence of a shared visual experience. The fact that signal universality was positively associated with communication success contributed to the greater success of gesture, compared to vocalization. Consistent with this finding, (hearing and non-signing) adults' success in interpreting sign language is predicted by the degree to which their gestures resemble the signs from sign language [66] (see also [67]), and children's success in using gesture to communicate is predicted by the degree to which their gestures resemble the gestures of others [50].

The greater universality of gesture, and its contribution to communication success, is predicted by an embodied view of cognition, which highlights the importance of the body to cognition [52,53]. Of particular relevance to the present study are embodied theories of language [54] and gesture production [55], which argue that our conceptual representations are grounded in perception and action. To the extent that people physically interact in similar environments, it follows that they will manually represent their environment in similar ways, giving rise to representations that are shared across people. This can help explain why participants from the same culture (and therefore similar environments) manually represented meaning in more similar ways than participants from a different culture (Experiment 1), and why severely vision-impaired participants represented meaning in the gesture modality in a similar way to sighted participants from the same culture (Experiment 2). For example, in Experiment 1, the word 'lock' was communicated similarly across cultures (by manually simulating turning a

key in a lock), but 'chain' was communicated differently: by manually simulating a pulling action (attached to something heavy) by an Australian Producer, and by manually simulating a throwing action (that represented a chain as an anchor) by a Ni-Vanuatu Producer. So, similarities and differences in how Producers use their body to interact with objects in their environment gave rise to similarities and differences in how meaning was embodied by Australian and Ni-Vanuatu Producers. By contrast, in the vocal modality, the opportunity for signal embodiment was absent. A variety of gesture and vocal example signals are available on the OSF: https://osf.io/36jpy/. Recognizing the greater affordance of gesture for embodied communication can help us more fully understand why communication success was twice as high for gesture than for (non-linguistic) vocalization across Experiments 1 and 2 (61.17% versus 29.04%).

Like other studies that simulate language evolution under controlled laboratory conditions (for reviews see [35,68–70]), the experimental results reported here rely on modern humans with modern brains, and mastery of at least one spoken language. Estimates of language origin range from 50 000 to half a million years ago [71]. While the human cognitive system is likely to have been subject to some genetic change since then, it is not clear why any feature that helped establish language would be discarded during the later evolution of the species [72]. This is the working assumption of researchers who use laboratory experiments to study language evolution. Furthermore, the pattern of results reported here is also observed among young children with less developed (albeit modern) brains and less mastery of language [49]. Similarly, naturalistic research has shown that deaf children raised in hearing households, but not exposed to sign language, develop similar rudimentary

systems of gestured communication (home-sign) in the absence of a language model [12]. We cannot know with any certainty if gesture was the primary modality for language creation among our prehistoric ancestors, but evidence from modern humans suggests a primary role for gesture during the first stages of language creation.

Although our results are consistent with a gesture-first theory of language origin, they do not rule out the possibility that gesture and speech together formed the first communication system. A key benefit of a multimodal theory of language origin is that it can account for the fact that all modern-day spoken languages have an integrated gesture–speech system [73,74]. However, experimental evidence for a multimodal origin is mixed: there is evidence that combining gesture with speech can improve communication efficiency [43], but there is also evidence that adding speech to gesture does not improve communication success [41] or can impede communication success [42]. These studies are limited by their reliance on WEIRD samples of participants and by their use of a small and potentially unrepresentative sample of stimuli. A valuable avenue for future research is to compare the communication success of the gesture, vocal and combined modalities across a range of different cultures (as per [40]) using a broad range of words (as per the present study).

## 5. Conclusion

We report two experiments that inform gesture-first and vocal-first theories of language origin. Using a task that prohibits the use of conventional language, communication success was twice as high for gestured signals than (non-linguistic) vocal signals. This was true within cultures and across cultures (Experiment 1), and even for severely vision-impaired participants (Experiment 2). These findings support a gesture-first theory of language origin. As predicted by an embodied account of human cognition (and by philosophers and early explorers), gestured signals were more *universal* than vocal signals. This was true within cultures, across cultures, and even for severely vision-impaired participants (despite the absence of a shared visual experience). Signal universality was positively correlated with communication success and so contributed to the success of gestured signals. The universality of

gesture means it is ideally suited to bootstrapping human communication among modern humans and therefore supports the hypothesis that gesture is the primary modality for language creation.

Ethics. Experiments 1 and 2 received approval from the University of Western Australia Ethics Committee. Participants read an information sheet before giving written consent to take part in the study. The information sheet and consent form (in braille for blind participants) were approved by the Ethics Committee. All methods were performed in accordance with the guidelines from the NHMRC/ARC/University Australia's National Statement on Ethical Conduct in Human Research.

Data accessibility. The data, R scripts, materials, electronic supplementary materials and example gestures/vocalizations are available on the OSF: https://osf.io/36jpy/.

Authors' contributions. N.F.: conceptualization, data curation, formal analysis, funding acquisition, investigation, methodology, project administration, supervision, visualization, writing—original draft and writing—review and editing; B.W.: conceptualization, investigation, methodology, project administration, software, writing—original draft and writing—review and editing; T.M.E.: conceptualization, investigation, methodology and writing—review and editing; Z.B.: investigation; N.D.K.: investigation; M.G.: investigation, methodology, writing—review and editing; C.J.L.: investigation; S.G.-M.: writing—original draft.

All authors gave final approval for publication and agreed to be held accountable for the work performed therein.

Competing interests. We declare we have no competing interests.

Funding. This research was supported by a University of Western Australia Research Collaboration Award (grant no. 12105105) awarded to authors N.F., C.J.L. and S.G.-M., and an ARC Centre of Excellence of the Dynamics of Language (Australian National University) Transdisciplinary and Innovation Grant awarded to authors N.F., T.M.E. and M.G.

Acknowledgements. We are deeply indebted to Catherine Regan and her late brother David Regan, who were essential to the recruitment of severely vision-impaired participants (Experiment 2). We are grateful to editor Robert Barton, and to Gerardo Ortega and two anonymous reviewers for their valuable feedback on an earlier version of the manuscript.

## Endnotes

[1] An Interpreter did not respond to one of the words presented, reducing the number of assessments of that word to 4.

[2] Producers skipped more trials in Experiment 2 than Experiment 1. This is probably due to the more complex word set used in Experiment 2.

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
