## [Peer Review File · Proceedings of the Royal Society B: Biological Sciences]

Review History

RSPB-2021-1116.R0 (Original submission)

Review form: Reviewer 1 (Gerardo Ortega)

Recommendation

Major revision is needed (please make suggestions in comments)

Scientific importance: Is the manuscript an original and important contribution to its field?

Good

General interest: Is the paper of sufficient general interest?

Excellent

Quality of the paper: Is the overall quality of the paper suitable?

Good

Is the length of the paper justified?

Yes

Should the paper be seen by a specialist statistical reviewer?

No

Do you have any concerns about statistical analyses in this paper? If so, please specify them explicitly in your report.

No

It is a condition of publication that authors make their supporting data, code and materials available - either as supplementary material or hosted in an external repository. Please rate, if applicable, the supporting data on the following criteria.

Is it accessible?

Yes

Is it clear?

Yes

Is it adequate?

Yes

Do you have any ethical concerns with this paper?

No

Comments to the Author

See attached pdf. (See Appendix A)

Review form: Reviewer 2

Recommendation

Accept with minor revision (please list in comments)

Scientific importance: Is the manuscript an original and important contribution to its field?

Excellent

General interest: Is the paper of sufficient general interest?

Good

Quality of the paper: Is the overall quality of the paper suitable?

Excellent

Is the length of the paper justified?

Yes

Should the paper be seen by a specialist statistical reviewer?

No

Do you have any concerns about statistical analyses in this paper? If so, please specify them explicitly in your report.

No

It is a condition of publication that authors make their supporting data, code and materials available - either as supplementary material or hosted in an external repository. Please rate, if applicable, the supporting data on the following criteria.

Is it accessible?

Yes

Is it clear?

Yes

Is it adequate?

Yes

Do you have any ethical concerns with this paper?

No

Comments to the Author

Generally, I liked the study, and the results were very interesting. The study overcomes some issues of previous work (e.g., culturally diverse sample, not only WEIRD participants), and it not only tries to investigate the naturalness of gestural communication, but also its universality. I have a few comments/questions for clarification.

Abstract Section

- I'd suggest to include the age of participants, as well as the cultures that were studied. Some readers skim the abstract for this information to see whether it is important/useful for their own work.

Introduction Section

- Line 55 onwards: The discussion of the great ape literature is interesting, however, it's also a bit over-simplistic and under-references. It would very much benefit from more detail and a more nuanced discussion of the evidence. For example, a range recent studies have shown that great apes are quite flexible in their use of vocal signals (e.g. work by Cathy Crockford and colleagues). When it comes to gestures, Bohn and colleagues have found that great apes struggle with the kind of gestures you are studying – iconic gestures. More broadly, the links between great ape gestural communication and human communication are complex and highly debated (see e.g. recent work by Fröhlich and Pika). Since you don't come back to this literature in the discussion, I was wondering if you might want to think about excluding this part completely. Instead, you could devote more space to the cross-cultural gesture literature.

- Line 77: It would be nice to mention the cultures involved in the study here. If you omit the great ape literature, you have some space to give a little bit more background about the cultures; especially why you compared those two? This would help a reader less familiar with cross-cultural work to better follow the introduction and the results.

Results Section

- I would move the details about the analysis to the methods section. They seem a bit out of place here.

- The results are difficult to understand without reading the methods. Line 100: 1-2 introductory sentences could be helpful (e.g., participants were asked to communicate a word by only using signs [...]. Producers could skip a trial [...])

- Please mention that the interpreters were all from Australia already in the results section. I think this is an important piece of information to interpret the results. One could argue that the study is not really a cross-cultural study because all the interpreters are Australian students. I'm not saying that this is the case and that this limits the merit of the study in any way – I just think it would be fair to mention this up front.

- It would be good to get some basic metrics on the different signals, especially how long they were. Maybe gestures were simply longer and therefore more informative and easier to understand (not that I think this is what's driving the effect, but would be interesting to discuss).

- It would be interesting to have a closer look at the different word types (e.g., are nouns or verbs easier to sign?)

- Figure 1: I highly recommend to change the color combinations. A red-green combination is always difficult for people with color blindness; especially Panel B and D are difficult from that perspective.

- Line 168: I would change "Blind Producers" to "Visually-Impaired Producers".

- Why are there so many more words used in Study 2 compared to Study 1? It might be worth to justify this – from a readers perspective it seems a bit odd.
- Figure 2: It would be interesting to also show the correlation between Similarity Rating and the Confidence Ratings.

Discussion

- Line 225-233: I would move this part to the introduction or at least mention it there. It opens up a whole new theory/explanation, and the reader would benefit from this background information before reading the results and the discussion.
- One of the main strengths of this study is the inclusion of two different cultures. The discussion part misses to address how differences between cultures might have affected the results. For instance, you point out cross-cultural differences in gesture production. While it is really interesting to see the videos as a direct comparison, I am missing an explanation for these differences.

Method Section

- Experiment 2 has a relatively low sample size. Is this the reason why there is this difference between the number of words used between Exp. 1 and 2?
- For clarity: Were the sighted participants allowed to wear glasses? Was the percentage of vision in the paper with or without any aids?
- I really appreciate that you made videos available in the supplementary material. However, there are only examples for nouns, I think readers would like to see how other word types were signed. Additionally, maybe 1-2 records for the vocalizations would be really interesting

Review form: Reviewer 3

Recommendation

Reject – article is not of sufficient interest (we will consider a transfer to another journal)

Scientific importance: Is the manuscript an original and important contribution to its field?

Marginal

General interest: Is the paper of sufficient general interest?

Marginal

Quality of the paper: Is the overall quality of the paper suitable?

Marginal

Is the length of the paper justified?

Yes

Should the paper be seen by a specialist statistical reviewer?

No

Do you have any concerns about statistical analyses in this paper? If so, please specify them explicitly in your report.

No

It is a condition of publication that authors make their supporting data, code and materials available - either as supplementary material or hosted in an external repository. Please rate, if applicable, the supporting data on the following criteria.

Is it accessible?

Yes

Is it clear?

Yes

Is it adequate?

Yes

Do you have any ethical concerns with this paper?

No

Comments to the Author

Overall summary:

The paper compared the capacity of gesture and vocal (non-verbal) communication to convey meaning information typically encoded by single words in English. The goal of the study was to determine whether gesture is more effective at conveying meaning than sound. Communication was tested across cultures and across groups with different visual experiences. Exp 1 collected gestures and vocalizations from a group of Ni-Vanuatu participant, a subsistence agriculturalists group, and a matched group of Western culture Australians. Australian college students were presented with the gestures and vocalizations and were asked to guess which of 16 English words each gesture/vocalization was depicting. The gestures/vocalizations were also rated for similarity across cultures. In Exp 2, a group of visually impaired and a group of sighted participants performed the gesturing and vocalization task and a group of sighted participants then guessed the meaning of gestures/vocalizations. The study reports that gestures were more interpretable within and across cultures as compared to vocalizations. The gestures were also more consistent across cultures than vocalizations, although still less similar across than within culture. Gesture similarity across people predicted how interpretable the gesture was within and across cultures. The paper concludes that gestures are a more natural and universal mode of communication and 'involved in the first steps of language creation.'

General comment:

The paper reports an interesting set of experimental tasks with a diverse range of participant groups. However, the writing does not make the interest of the study clear and does not describe the methods in sufficient detail to make the contribution of significant value. The conclusions of the Discussion are not clearly tied to the literature and highly speculative.

Introduction:

1. The introduction of the paper does not sufficiently motivate the study.

The theoretical grounding and motivation of the experiment is not clear. The original question 'naturalness' of gesture is not clearly defined. What does it really mean for gesture to be natural or not? Relative to what? What does it mean for something to be 'natural?' Is this really a solid scientific/philosophical concept? If so, it needs to be clearly defined. The introduction of the paper mentions the affordance of gesture to communicate meaning without really discussing why this might be. It is of course possible for gesture to be iconic in a way that vocal communication is not. Is this relevant? What are the relevant differences between gesture and vocal communication as information transfer mediums? It's also worth noting that the 'gesture first' theory is not directly empirically testable, since it asks about the evolutionary origins of gesture.

2. The empirical motivation of the experiment is also not clear. The introduction states that previous studies have been done with small, homogenous samples of participants and small samples of meanings. This is fine but it doesn't say how a better sample would resolve any questions. Nor does it motivated the current samples.

3. The paper lacks sufficient motivational explanation for the specific groups tested. Why blind participants? The logic needs to be spelled out.

4. Universality is a descriptive concept. Gestures are more similar, but it doesn't really say why this is the case.

Methods:

5. The methods are quite difficult to follow and contain insufficient information. Important information is either omitted or placed parenthetically and therefore difficult to find. Below are some detailed examples but this is a general issue in the paper.

6. Basic information about the cultures/groups being studied is not provided early in the methods. What was the native language of the participants? In one place it says the instructions/ words were translated into Bislama but it is not stated whether this was the native language for all the participants. Were the words chosen also frequent in Bislama? Did all the words chosen correspond to single lexemes in Bislama? What was done to ensure that words were familiar in each culture? In the results there is less gesture similarity among Ni-Vanuatu participants than Australian participants. This raises the worry that the materials were less accessible to the Ni-Vanuatu group. Earlier in the text it also says that some English words were omitted because they were unfamiliar to the Ni-Vanuatu participants. Are the English words 'adjective' 'noun' and 'verb' familiar to the Ni-Vanuatu participants?

7. The blind group is small, heterogenous and insufficiently described. Some are totally blind, some have significant vision. Only 5 of the blind participants are congenitally blind. Why is it relevant to test late blind participants in this particular study? Cause of vision loss is not reported. Are all participants Native English speakers? Are participants screened in any way for other cognitive disabilities?

8. Education matching is overly course across groups and insufficient education information is reported.

9. Number of trials skipped should be reported by group. While the mean is low, the SD is high.

10. Insufficient information is provided about the tested groups. What relevant information do we need to know about Vanuatu participants? The paper states that these participants are subsistence farmers but that they are there to record a music CD. What does this mean? It says the participants have had up to 6 years of formal education. What is the average education? Are these literate participants? Are they being tested by a native speaker? In what language?

11. The word sampling procedure is interesting and a nice touch to the experiments.

12. Exp 2, some participants are skipping a very large proportion of words. $M=22\%$; $SD=42\%$. 1) Please report information by group and by condition. Are there group differences? 2) What distinguishes the skipped from the un-skipped words? How does this affect the conclusions? It seems possible that gesture is bad at conveying many high frequency words and these are being skipped.

13. I could find no description of how similarity data were collected. Who rated the gestures? Was it the same 'interpreters' who did the meaning matching? Should we think of these data as independent? How reliable are the measures? What criteria are being used?

Results

14. Sign similarity data from blind participants is heterogenous and it is not clear how this is related to blindness onset or severity.

15. In Exp 2, there is no effect of similarity on communication success for gesture when collapsing across groups and the effect is larger for vocalizations. Why is this? (lines 187-190)
Discussion

16. "The greater affordance of the manual-visual modality for iconic sign production is a common explanation for why gesture is more successful at bootstrapping human communication than vocalization [30,43,44]"

The issue of iconicity should be raised in the introduction. But it is not at all clear to me that 'universality' and iconicity are not the same thing here. Gesture may be more universal precisely because it lends itself more to iconicity. As noted in my comments on the Introduction, universality is also a descriptive rather than an explanatory construct. All it says is that something the same across people, but it doesn't say why.

17. The embodied cognition explanation of the effects is a stretch in a variety of ways. It's not clear how it relates to specific theories of embodiment. It also doesn't seem to have any clear connection to the data in the experiment or data from prior work. (line 234)

18. Likewise, the claim that people in the same culture use similar gesture because they live in a more similar environment highly speculative. There are many other alternative explanations. For example, people living in the same culture learn conventional gestures of that culture. This cannot be ruled out for blind speakers either. Take for example curse gestures, these vary across cultures and are conventionalized. I very much doubt that blind Americans don't know what it means to show the middle finger to someone. That is not to say that common environment or instruments do not play a role. Simply that there is no evidence one way or the other.

19. "In the vocal modality the opportunity for embodiment is absent or greatly diminished" I don't understand what this sentence means. How are sounds or actions inherently embodied? The term 'embodied' is being used very loosely.

20. The last paragraph of the Discussion does not seem to connect to the rest of the paper. Relevance is not clear. Relationship to language is not clear.

21. The end of the last paragraph of the Discussion is wildly speculative. "Note that our results make a convincing case that gesture is likely to have been involved in the first steps of language creation." The present results cannot speak to this issue. No information about language and gesture evolution presented in the paper.

Decision letter (RSPB-2021-1116.R0)

19-Jul-2021

Dear Dr Fay:

I am writing to inform you that your manuscript RSPB-2021-1116 entitled "Gesture is a Natural Modality for Language Creation" has, in its current form, been rejected for publication in Proceedings B.

This action has been taken on the advice of referees, who have recommended that substantial revisions are necessary. With this in mind we would be happy to consider a resubmission, provided the comments of the referees are fully addressed. However please note that this is not a provisional acceptance.

Sincerely,
 Dr Robert Barton
 mailto: proceedingsb@royalsociety.org

Associate Editor
 Comments to Author:

This is an interesting paper testing the gestural origins of language by comparing the creation of novel gestural or vocal signs, and their efficacy in communication. It uses rarely or difficult-to-study populations to address key questions about the emergence of human language. The inclusion of non-WEIRD sample as an especially important component of the work. However, the paper would also benefit from greater theoretical nuance in setting up the questions, as well as a more careful presentation of the methods and results to make sure these aspects are clear to readers.

On important issue raised by reviewers concerns greater theoretical nuance in the introduction, such as in defining and using the terms naturalness and universality (R1,R3). If these terms are going to be used, they should be used with much greater care. Similarly, while the inclusion of different groups is a strength of a paper, their inclusion needs to be better justified. What does the inclusion of blind participants tell us about language evolution and/or the gestural origins of language, for example – an issue raised by both R1 and R3 as well. There are also important comments about the description of the methods and participant groups raised by all three reviewers, which is important for readers to understand and interpret the paper. Finally, an important point is to make sure this paper is accessible and relevant to a broader biological readership. In this sense, I think that discussion of comparative work is important, as an evolutionary lens on this study will help contextualize it for such an audience, but you should attend to the comments from R2 concerning findings from this field and generally the manner in which this work is cited and discussed. More generally, this means a revision of the discussion, which as several reviewers also mention is fairly speculative and does not seem to return to question about language origins and evolution raised in the introduction.

Reviewer(s)' Comments to Author:

Referee: 1

Comments to the Author(s)

See attached pdf

Referee: 2

Comments to the Author(s)

Generally, I liked the study, and the results were very interesting. The study overcomes some issues of previous work (e.g., culturally diverse sample, not only WEIRD participants), and it not only tries to investigate the naturalness of gestural communication, but also its universality. I have a few comments/questions for clarification.

Abstract Section

- I'd suggest to include the age of participants, as well as the cultures that were studied. Some readers skim the abstract for this information to see whether it is important/useful for their own work.

Introduction Section

- Line 55 onwards: The discussion of the great ape literature is interesting, however, it's also a bit over-simplistic and under-references. It would very much benefit from more detail and a more nuanced discussion of the evidence. For example, a range recent studies have shown that great apes are quite flexible in their use of vocal signals (e.g. work by Cathy Crockford and colleagues). When it comes to gestures, Bohn and colleagues have found that great apes struggle with the kind of gestures you are studying - iconic gestures. More broadly, the links between great ape gestural communication and human communication are complex and highly debated (see e.g. recent work by Fröhlich and Pika). Since you don't come back to this literature in the discussion, I was wondering if you might want to think about excluding this part completely. Instead, you could devote more space to the cross-cultural gesture literature.

- Line 77: It would be nice to mention the cultures involved in the study here. If you omit the great ape literature, you have some space to give a little bit more background about the cultures; especially why you compared those two? This would help a reader less familiar with cross-cultural work to better follow the introduction and the results.

Results Section

- I would move the details about the analysis to the methods section. They seem a bit out of place here.

- The results are difficult to understand without reading the methods. Line 100: 1-2 introductory sentences could be helpful (e.g., participants were asked to communicate a word by only using signs [...]. Producers could skip a trial [...])

- Please mention that the interpreters were all from Australia already in the results section. I think this is an important piece of information to interpret the results. One could argue that the study is not really a cross-cultural study because all the interpreters are Australian students. I'm not saying that this is the case and that this limits the merit of the study in any way - I just think it would be fair to mention this up front.

- It would be good to get some basic metrics on the different signals, especially how long they were. Maybe gestures were simply longer and therefore more informative and easier to understand (not that I think this is what's driving the effect, but would be interesting to discuss).

- It would be interesting to have a closer look at the different word types (e.g., are nouns or verbs easier to sign?)

- Figure 1: I highly recommend to change the color combinations. A red-green combination is always difficult for people with color blindness; especially Panel B and D are difficult from that perspective.

- Line 168: I would change "Blind Producers" to "Visually-Impaired Producers".

- Why are there so many more words used in Study 2 compared to Study 1? It might be worth to justify this - from a readers perspective it seems a bit odd.

- Figure 2: It would be interesting to also show the correlation between Similarity Rating and the Confidence Ratings.

Discussion

- Line 225-233: I would move this part to the introduction or at least mention it there. It opens up a whole new theory/explanation, and the reader would benefit from this background information before reading the results and the discussion.

- One of the main strengths of this study is the inclusion of two different cultures. The discussion part misses to address how differences between cultures might have affected the results. For instance, you point out cross-cultural differences in gesture production. While it is really interesting to see the videos as a direct comparison, I am missing an explanation for these differences.

Method Section

- Experiment 2 has a relatively low sample size. Is this the reason why there is this difference between the number of words used between Exp. 1 and 2?

- For clarity: Were the sighted participants allowed to wear glasses? Was the percentage of vision in the paper with or without any aids?

- I really appreciate that you made videos available in the supplementary material. However, there are only examples for nouns, I think readers would like to see how other word types were signed. Additionally, maybe 1-2 records for the vocalizations would be really interesting

Referee: 3

Comments to the Author(s)

Overall summary:

The paper compared the capacity of gesture and vocal (non-verbal) communication to convey meaning information typically encoded by single words in English. The goal of the study was to determine whether gesture is more effective at conveying meaning than sound. Communication was tested across cultures and across groups with different visual experiences. Exp 1 collected gestures and vocalizations from a group of Ni-Vanuatu participant, a subsistence agriculturalists group, and a matched group of Western culture Australians. Australian college students were presented with the gestures and vocalizations and were asked to guess which of 16 English words each gesture/vocalization was depicting. The gestures/vocalizations were also rated for similarity across cultures. In Exp 2, a group of visually impaired and a group of sighted participants performed the gesturing and vocalization task and a group of sighted participants then guessed the meaning of gestures/vocalizations. The study reports that gestures were more interpretable within and across cultures as compared to vocalizations. The gestures were also more consistent across cultures than vocalizations, although still less similar across than within culture. Gesture similarity across people predicted how interpretable the gesture was within and across cultures. The paper concludes that gestures are a more natural and universal mode of communication and 'involved in the first steps of language creation.'

General comment:

The paper reports an interesting set of experimental tasks with a diverse range of participant groups. However, the writing does not make the interest of the study clear and does not describe the methods in sufficient detail to make the contribution of significant value. The conclusions of the Discussion are not clearly tied to the literature and highly speculative.

Introduction:

1. The introduction of the paper does not sufficiently motivate the study.

The theoretical grounding and motivation of the experiment is not clear. The original question 'naturalness' of gesture is not clearly defined. What does it really mean for gesture to be natural or not? Relative to what? What does it mean for something to be 'natural'? Is this really a solid scientific/philosophical concept? If so, it needs to be clearly defined. The introduction of the paper mentions the affordance of gesture to communicate meaning without really discussing

why this might be. It is of course possible for gesture to be iconic in a way that vocal communication is not. Is this relevant? What are the relevant differences between gesture and vocal communication as information transfer mediums? It's also worth noting that the 'gesture first' theory is not directly empirically testable, since it asks about the evolutionary origins of gesture.

2. The empirical motivation of the experiment is also not clear. The introduction states that previous studies have been done with small, homogenous samples of participants and small samples of meanings. This is fine but it doesn't say how a better sample would resolve any questions. Nor does it motivated the current samples.

3. The paper lacks sufficient motivational explanation for the specific groups tested. Why blind participants? The logic needs to be spelled out.

4. Universality is a descriptive concept. Gestures are more similar, but it doesn't really say why this is the case.

Methods:

5. The methods are quite difficult to follow and contain insufficient information. Important information is either omitted or placed parenthetically and therefore difficult to find. Below are some detailed examples but this is a general issue in the paper.

6. Basic information about the cultures/groups being studied is not provided early in the methods. What was the native language of the participants? In one place it says the instructions/ words were translated into Bislama but it is not stated whether this was the native language for all the participants. Were the words chosen also frequent in Bislama? Did all the words chosen correspond to single lexemes in Bislama? What was done to ensure that words were familiar in each culture? In the results there is less gesture similarity among Ni-Vanuatu participants than Australian participants. This raises the worry that the materials were less accessible to the Ni-Vanuatu group. Earlier in the text it also says that some English words were omitted because they were unfamiliar to the Ni-Vanuatu participants. Are the English words 'adjective' 'noun' and 'verb' familiar to the Ni-Vanuatu participants?

7. The blind group is small, heterogenous and insufficiently described. Some are totally blind, some have significant vision. Only 5 of the blind participants are congenitally blind. Why is it relevant to test late blind participants in this particular study? Cause of vision loss is not reported. Are all participants Native English speakers? Are participants screened in any way for other cognitive disabilities?

8. Education matching is overly course across groups and insufficient education information is reported.

9. Number of trials skipped should be reported by group. While the mean is low, the SD is high.

10. Insufficient information is provided about the tested groups. What relevant information do we need to know about Vanuatu participants? The paper states that these participants are subsistence farmers but that they are there to record a music CD. What does this mean? It says the participants have had up to 6 years of formal education. What is the average education? Are these literate participants? Are they being tested by a native speaker? In what language?

11. The word sampling procedure is interesting and a nice touch to the experiments.

12. Exp 2, some participants are skipping a very large proportion of words. $M=22\%$; $SD=42\%$. 1) Please report information by group and by condition. Are there group differences? 2) What distinguishes the skipped from the un-skipped words? How does this affect the conclusions? It

seems possible that gesture is bad at conveying many high frequency words and these are being skipped.

13. I could find no description of how similarity data were collected. Who rated the gestures? Was it the same 'interpreters' who did the meaning matching? Should we think of these data as independent? How reliable are the measures? What criteria are being used?

Results

14. Sign similarity data from blind participants is heterogenous and it is not clear how this is related to blindness onset or severity.

15. In Exp 2, there is no effect of similarity on communication success for gesture when collapsing across groups and the effect is larger for vocalizations. Why is this? (lines 187-190)

Discussion

16. "The greater affordance of the manual-visual modality for iconic sign production is a common explanation for why gesture is more successful at bootstrapping human communication than vocalization [30,43,44]"

The issue of iconicity should be raised in the introduction. But it is not at all clear to me that 'universality' and iconicity are not the same thing here. Gesture may be more universal precisely because it lends itself more to iconicity. As noted in my comments on the Introduction, universality is also a descriptive rather than an explanatory construct. All it says is that something the same across people, but it doesn't say why.

17. The embodied cognition explanation of the effects is a stretch in a variety of ways. It's not clear how it relates to specific theories of embodiment. It also doesn't seem to have any clear connection to the data in the experiment or data from prior work. (line 234)

18. Likewise, the claim that people in the same culture use similar gesture because they live in a more similar environment highly speculative. There are many other alternative explanations. For example, people living in the same culture learn conventional gestures of that culture. This cannot be ruled out for blind speakers either. Take for example curse gestures, these vary across cultures and are conventionalized. I very much doubt that blind Americans don't know what it means to show the middle finger to someone. That is not to say that common environment or instruments do not play a role. Simply that there is no evidence one way or the other.

19. "In the vocal modality the opportunity for embodiment is absent or greatly diminished" I don't understand what this sentence means. How are sounds or actions inherently embodied? The term 'embodied' is being used very loosely.

20. The last paragraph of the Discussion does not seem to connect to the rest of the paper. Relevance is not clear. Relationship to language is not clear.

21. The end of the last paragraph of the Discussion is wildly speculative. "Note that our results make a convincing case that gesture is likely to have been involved in the first steps of language creation." The present results cannot speak to this issue. No information about language and gesture evolution presented in the paper.

Author's Response to Decision Letter for (RSPB-2021-1116.R0)

See Appendix B.

RSPB-2022-0066.R0

Review form: Reviewer 1

Recommendation

Major revision is needed (please make suggestions in comments)

Scientific importance: Is the manuscript an original and important contribution to its field?

Excellent

General interest: Is the paper of sufficient general interest?

Excellent

Quality of the paper: Is the overall quality of the paper suitable?

Excellent

Is the length of the paper justified?

Yes

Should the paper be seen by a specialist statistical reviewer?

Yes

Do you have any concerns about statistical analyses in this paper? If so, please specify them explicitly in your report.

Yes

It is a condition of publication that authors make their supporting data, code and materials available - either as supplementary material or hosted in an external repository. Please rate, if applicable, the supporting data on the following criteria.

Is it accessible?

No

Is it clear?

N/A

Is it adequate?

N/A

Do you have any ethical concerns with this paper?

No

Comments to the Author

See attached pdf. (See Appendix C)

Decision letter (RSPB-2022-0066.R0)

09-Feb-2022

Dear Dr Fay

I am pleased to inform you that your manuscript RSPB-2022-0066 entitled "Gesture is the Primary Modality for Language Creation" has been accepted for publication in Proceedings B.

The referee(s) have recommended publication, but also suggest some minor revisions to your manuscript. Therefore, I invite you to respond to the referee(s)' comments and revise your manuscript. Because the schedule for publication is very tight, it is a condition of publication that you submit the revised version of your manuscript within 7 days. If you do not think you will be able to meet this date please let us know.

In order to ensure effective and robust dissemination and appropriate credit to authors the dataset(s) used should be fully cited. To ensure archived data are available to readers, authors should include a 'data accessibility' section immediately after the acknowledgements section.

This should list the database and accession number for all data from the article that has been made publicly available, for instance:

[http://datadryad.org/submit?journalID=RSPB&manu=\(Document not available\)](http://datadryad.org/submit?journalID=RSPB&manu=(Document%20not%20available)) which will take you to your unique entry in the Dryad repository. If you have already submitted your data to dryad you can make any necessary revisions to your dataset by following the above link. Please see <https://royalsociety.org/journals/ethics-policies/data-sharing-mining/> for more details.

Sincerely,

Dr Robert Barton

Associate Editor

Board Member

Comments to Author:

I have read the revision as well as the response to the reviewer's comments, and I think that this is a responsive revision that does a nice job addressing the major points raised in response to the prior version of the paper. The reviewer has several comments that should be addressed in a revision. I would highlight especially two of these points: (1) data accessibility and (2) inclusion of a comment about how the multiple-choice task, which thus constrained potential interpretations of the communicative acts, may have impacted performance would be useful.

In addition, I have several smaller comments I would like to see addressed, several of which have to do with toning down the wording of claims:

- The last line of the abstract should be edited, for example to read something like "Our results support the hypothesis that gesture is the primary modality for language creation." The abstract and paper situates these studies in terms of big-picture questions concerning the historical origins of language, including in evolutionary time, and the experiments cannot adjudicate these historical questions in the way the current wording suggests (which is discussed also in the discussion section).
- Similar comment concerning the conclusion section line 463: edit to "and therefore supports the hypothesis that gesture is the primary modality for language creation."
- Introduction line 46: I would suggest removing the word "ambitious."
- Line 33: This wording incorrectly suggests that it is not possible to compare communication across modalities in general in nonhumans. I don't think this is accurate even if there is some debate about specifically comparing meaning of communicative acts.
- Line 71: Would suggest an edit to "The experiments reported here aim to overcome these issues."

Reviewer(s)' Comments to Author:
Referee: 1
Comments to the Author(s).
See attached pdf.

Author's Response to Decision Letter for (RSPB-2022-0066.R0)

See Appendix D.

Decision letter (RSPB-2022-0066.R1)

14-Feb-2022

Dear Dr Fay

I am pleased to inform you that your manuscript entitled "Gesture is the Primary Modality for Language Creation" has been accepted for publication in Proceedings B.

Your article has been estimated as being 9 pages long. Our Production Office will be able to confirm the exact length at proof stage.

Data Accessibility section

Open Access

Paper charges

All supplementary materials accompanying an accepted article will be treated as in their final form. They will be published alongside the paper on the journal website and posted on the online

figshare repository. Files on figshare will be made available approximately one week before the accompanying article so that the supplementary material can be attributed a unique DOI.

Sincerely,
Editor, Proceedings B
<mailto:proceedingsb@royalsociety.org>

Appendix A

Review

In this study the authors investigated whether gesture or vocalisations were more efficient in expressing meaning in the absence of a conventionalised language. Participants were individuals from a WEIRD culture (Australia) and Vanuatu (experiment 1) and sighted and blind participants (experiment 2). In both experiments, producers were asked to generate gestures or vocalisations for a set of concepts, which then in turn were shown to other participants who had to guess their meaning (selecting an option from a list of 16 words). Results show that participants were more accurate at interpreting gestures than vocalisation and this result extended across populations. The results also show that gestures were rated more similar across participants than vocalisations. This pattern is observed across groups. The authors argue that gesture is universal and thus this facilitates comprehension across groups.

This is a very interesting study which adds to the on-going debate about the origins of language. This is a very important piece of work in that it tests and compares non-WEIRD cultures and their ability to interpret gestures and signs. The design and the statistical analysis of the study are impeccable and the interpretation of the results is clever and insightful. I think this study can be of interest to scientists interested in language evolution but also researchers studying gesture, embodiment, sign languages and iconicity. Before the manuscript is endorsed for publication, the authors may want to address the following issues.

1) Naturalness. It is not clear what the authors mean by this word. It's a very strong and loaded term but a rather ambiguous one. Also, it's not clear how it differs from 'universality'. The authors may want to explain what they mean by naturalness, and importantly, how other forms of communication (e.g. vocalisation) are less natural.

2) Universality. One of the most important issues that need to be addressed is that the authors do not explain the features that make gesture universal. For example, the authors argue that the universality of gesture allows people to communicate different concepts. I would like to encourage the authors to spell out this point and explain in more detail how gesture can achieve this feat. There are different types of gestures (iconic, beat, deictic) so it would be important to explain what types of gestures are most efficient to convey meaning.

3) Multimodality. '... vocalisation can ground shared meaning, but communication is more successful in gesture' (page 4). There is conflicting evidence to support this claim because some studies actually show that vocalisations can be understood across speakers of different languages (Ćwiek et al., 2021). In addition, there is also the option of speech AND gesture (multimodal). The findings presented in this paper don't support the notion that perhaps the origins of language were multimodal. Why only gesture and not gesture *and* speech at the same time (after all, nowadays we use gesture in high synchrony with speech). I'd like to invite the authors to discuss in more detail why should rule out a multimodal-first approach to language. This point is relevant because some could argue that in the actual experiment a third condition was missing: vocalisation+speech. In fact, as noted in the conclusion of the manuscript, recent studies that suggest that higher accuracy can be achieved in multimodal utterances (Machuc Silva et al 2020).

4) Blind participants. I think that one of the strengths of the paper is that the authors tested a group of blind participants. This is a very important contribution and one that could not have

been easy to achieve. That said, while blind people are a very interesting test case I feel that their inclusion in the study is not well justified. What can blind participants tell us about language evolution and the preference for gesture at the origins of language emergence? The authors should motivate more the use of this experimental group.

5) Multiple choice task. Communicative success was measured by participants selecting the correct response out of 16 options (page 13). I think that the authors should provide a strong justification of why this is the best possible way to measure communicative success. When participants are presented with a list of options, they are selecting the most plausible choice; they are not making free form-meaning associations. I agree that a multiple choice facilitates data collection but it doesn't really support the notion of 'universality' as the authors argue.

6) The authors argue that gestures are more accurately understood than vocalisation and that this finding aligns with research showing that similarity between gesture and sign also facilitates sign interpretation (page 10). Sign languages and gestures are different communicative systems so perhaps the authors may want to elaborate on how their findings align with this research. In fact, (Ortega, Schiefner, & Özyürek, 2019) argue that gesture helps the interpretation of signs when they represent bodily actions (e.g., the action of swimming or eating with a spoon). They make similar claims about production and interpretation of iconic gestures that resemble actions (Ortega & Ozyurek, 2019). This is in line with the authors discussion of embodied theories of language and cognition. Perhaps the authors may want to link gestures' 'universality' with action-based manual representations.

Minor comments

'... greater universality'. What does this mean? This is a very loaded term that can lend itself to misinterpretation.

'...communicate meanings across minds' (page 10). What do the authors mean by minds? I interpret this as being telepathy! This should be re-phrased to 'communicate across participants'.

'gestured signs relative to vocal signs'. Perhaps the authors may want to reconsider the use of the word 'sign'. These could be confused with signs from conventionalised sign languages of deaf communities. Symbols? Labels? The use of 'signs' becomes a bit confusing when the authors start referring to research on sign language research.

'sign universality' (page 8 and throughout the paper). This is a very misleading label. I would suggest that the authors use a different word. In the Results section, for instance, 'sign universality' actually refers to degree of similarity. I am not sure it is possible to talk about 'universality' when only two cultures were tested.

'...gestured signs are more universal than vocal signs'. I would strongly encourage the authors to refrain from using the word 'universal' or they should explain in more detail the features that makes gestures universal. Importantly, they should also be explicit about why vocalisations are NOT universal. Recent work has shown that vocalisations can successfully be interpreted by participants from various cultures (Ćwiek et al., 2021). What makes one but not the other universal?

Appendix B

THE UNIVERSITY OF
WESTERN AUSTRALIA
Achieving International Excellence

School of Psychological Science
35 Stirling Highway, CRAWLEY
Perth, Western Australia, WA 6009
T +61 (0)8 6488 2688
F +61 (0)8 6488 1006
E nicolas.fay@gmail.com
CRICOS Provider Code: 00126G

13 January 2022

The Editor, *Proceeding of the Royal Society B*

Dear Professor Barton,

Thank you for overseeing the review of our manuscript. We are encouraged by the positive assessment it received from the reviewers¹, and welcome the revisions they suggested. The manuscript has been thoroughly revised in light of the reviewer comments, and is much improved. We are grateful to the reviewers for their valuable feedback. Our response is organised around the sections of the report.

Abstract

1. **R2** requested that we include the age of the participants in the abstract.

This has been done.

Introduction

2. As summarised by the Editor, there was a consensus among the reviewers that the introduction would benefit from greater theoretical nuance. In particular, the experiments should be more clearly motivated, a more thorough review of comparative work be provided, and use of the terms 'naturalness' and 'universality' be reconsidered.

The introduction has been thoroughly revised. This includes a more thorough review of the comparative work (**R2**) (lines 47-59; Note that line numbers apply to the clean manuscript). This was valuable because: 1) adding the comparative evidence in favour of a vocal-first account (in addition to the already cited gesture-first evidence) made the paper's research question more compelling, and 2) it highlighted the current inability of comparative work to directly compare across the gesture and vocal modalities (a strength of the current study).

1 We were especially encouraged by the strong endorsement provided by **R1**, "*This is a very interesting study which adds to the on-going debate about the origins of language. This is a very important piece of work in that it tests and compares non-WEIRD cultures and their ability to interpret gestures and signs. The design and the statistical analysis of the study are impeccable and the interpretation of the results is clever and insightful. I think this study can be of interest to scientists interested in language evolution but also researchers studying gesture, embodiment, sign languages and iconicity*" and by **R2**, "*Generally, I liked the study, and the results were very interesting. The study overcomes some issues of previous work (e.g., culturally diverse sample, not only WEIRD participants), and it not only tries to investigate the naturalness of gestural communication, but also its universality*".

Stronger motivation for Experiment 1 and Experiment 2 is provided. A separate paragraph now outlines the rationale for each study. Stronger motivation for Experiment 1 has been added by discussing recent research showing the greater *salience* of gestured signals compared to vocal signs (in addition to, but separate from, the tendency for gestures to be more iconic than vocal signals) and the positive correlation between signal salience and communication success (lines 73-90). This discussion also justifies our use of the term ‘universality’ and its operationalisation as signal similarity (**R1** and **R3**). We agree with **R1** that ‘naturalness’ is a loaded and ambiguous term (see also **R3**), so no longer use this term. Stronger motivation for Experiment 2 is provided by framing Experiment 2 as a test of *why* gesture is likely to be a more universal means of communication than non-linguistic vocalisation (**R1** and **R3**) (lines 105-120). We cite embodied theories of cognition to motivate the prediction that gesture will be more universal than vocalisation (in response to **R1**, and as recommended by **R2** who suggested we move our coverage of this literature from the discussion section to the introduction).

3. **R1** suggested we use the term ‘signal’ instead of ‘sign’ as sign could be confused with sign language.

This is an excellent suggestion, and has been implemented throughout the manuscript.

Methods

4. As summarised by the Editor, there was a consensus among the reviewers that the paper would benefit from a more careful presentation of the methods.

The method sections have been revised in light of the reviewer feedback. A fundamental change is that the method sections now precede the results sections (as recommended by **R2**). This change will help the reader interpret and comprehend the results. We respond to the other points raised by the reviewers below.

5. **R1** requested that we justify the MCQ approach to measuring communication success.

This has been done (lines 192-199).

6. **R2** suggested that we move the analytic details to the methods section.

This has been done (lines 183-228). We now justify each dependent variable (as per point 5) and the analytic approach. Doing so allowed us to streamline the results sections.

7. **R2** requested that we mention that the Interpreters were all from Australia.

This is now explicit in multiple places in the methods sections.

8. **R2** asked why there were so many more words used in Experiment 2 than Experiment 1, and noted that this should be justified (otherwise it might seem odd from a reader’s perspective).

This is explained in the methods section of Experiment 1 (lines 146-151). We have also added a note to the methods section of Experiment 2 stating, “*Sampling Producers from the same culture allowed us to use a much larger corpus of words compared to Experiment 1 (where we sampled a smaller corpus of basic words to ensure their meanings were shared across cultures).*” (lines 254-256).

9. **R2** suggested we change the label “Blind Producers” to “Visually-Impaired Producers”.

This is an excellent suggestion, which has been implemented (we now refer to Vision-Impaired Producers).

10. **R2** notes that Experiment 2 has a relatively low sample size, and asks if this is the reason why there is a difference between the number of words used in Experiment 1 and 2?

The reason for the low sample size in Experiment 2 is that severely vision-impaired Producers were exceptionally difficult to recruit. In fact, we were only able to recruit vision-impaired participants because one of our students had a blind brother. The smaller sample size (relative to Experiment 1) is now explained in the methods section of Experiment 2 (lines 235-237). Participant sample size is unrelated to the number of words sampled in Experiment 1 and 2 (see point 8).

11. **R3** requested that we note the basic language of the Ni-Vanuatu participants.

This has been done (line 158-159).

12. **R3** asked if the words chosen in Experiment 1 (180 basic English words) were frequent in Bislama, and if they correspond to single lexemes in Bislama. Relatedly, **R3** asked what was done to ensure the words were familiar in each culture.

R3's concern that the materials may have been less accessible to the Ni-Vanuatu participants is well-taken. Indeed, and as noted, pilot testing indicated that lower frequency words (from the Corpus of Contemporary American English) were often unfamiliar to the Ni-Vanuatu participants, hence our decision to create and use a corpus of 180 basic English words in Experiment 1 (lines 146-151).

We believe the opportunity to skip trials may have mitigated any bias in the materials used (e.g., Ni-Vanuatu participants could skip words they were unfamiliar with). This has now been made explicit in the manuscript (lines 184-190). The fact that, on average, very few trials were skipped by the Ni-Vanuatu participants suggests they were familiar with the words included. In any case, the study reported is primarily concerned with differences in performance across modalities as opposed to performance differences across cultures.

13. **R3** requested further information on the educational characteristics of the Ni-Vanuatu group.

This point is well-taken. Unfortunately we have no further information on the educational profile of the Ni-Vanuatu group. Education was a sensitive topic among this group, so we were reluctant to probe more deeply.

14. **R3** requested further information on the Ni-Vanuatu group, including clarity around their visit to Vanuatu to record a music CD, the language used to conduct the experiment and if they were being tested by a native speaker.

We have added further clarification by noting that several of the participants from Pentecost Island were members of a band who were visiting Vanuatu to record a music CD (lines 136-137). We now explicitly state that the experiment was run in Bislama, by the translator (author MG) (lines 158-160).

15. **R3** requested further information on the vision-impaired group, such as the cause of vision loss, their native language and if we screened for cognitive disabilities.

We have no further information on the cause of vision loss for the 5 participants who are not congenitally blind. We have added that their native language is English (line 235). No cognitive disabilities were reported, but this was not tested for (in any of the groups recruited).

16. **R3** requested further information about the similarity rating procedure, specifically who rated the pairs of signals for similarity and if the ratings are reliable.

This is an important point that is now clarified in the paper (lines 211-214). The similarity ratings were made by author CL. A subset of the data (20%) was rated by author BW, and an inter-rater reliability test indicated substantial agreement between the raters (Krippendorff's alpha = .714).

Results

R1 noted that, "*The design and the statistical analysis of the study are impeccable and the interpretation of the results is clever and insightful*". Consistent with this, the reviewers did not identify any significant issues in the results section, but did make several valuable suggestions.

17. **R2** recommended that we reconsider the colour combinations used in the figures as they may present difficulties to people with colour blindness.

This is a valuable point. The figures have been reformatted using a colourblind-friendly palette.

18. **R2** requested that we mention that the Interpreters were all from Australia in the results section.

This is done (lines 298-299).

19. **R2** notes that it would be good to get some basic metrics on the different signals, especially how long they were. **R2** wondered if the gestures were simply longer and therefore more informative and easier to understand (noting that, "not that I think this is what's driving the effect, but would be interesting to discuss").

As **R2** suspects, gestured signals were longer in duration than vocal signals. For example, in Experiment 1 the mean duration of gestured signals was 3.19 seconds (Australian Producers) and 3.84 seconds (Vanuatu Producers), whereas the mean duration of vocal signals was 1.98 seconds (Australian Producers) and 1.88 seconds (Vanuatu Producers). However, in each case there was no statistical evidence of a correlation between signal duration and communication success ($p < .360$; see below Figure). We therefore conclude that the longer duration of the gestured signals does not explain their communication success.

20. **R2** notes that it would be interesting to have a closer look at the different word types, e.g., are nouns or verbs easier to sign?

In the gesture modality verbs were communicated more successfully than nouns and nouns were communicated more successfully than adjectives: 78%, 74%, 60% for Australian Producers and 64%, 55%, 46% for Ni-Vanuatu Producers. The pattern was less clear in the vocal modality: 33%, 31%, 34% for Australian Producers and 16%, 14%, 13% for Ni-Vanuatu Producers. Importantly, and consistent with the paper's key message, gesture was substantially more successful than vocalization for each word type.

Because we have no specific hypothesis with regard to word type, we prefer not to include this information in the manuscript, especially given the tight space constraints at Proc B. We note that our data are available on the OSF, so those interested in exploring this aspect of the data can do so.

21. **R2** noted that it would be interesting to show the correlation between the signal similarity ratings and the confidence ratings.

The Interpreter confidence ratings are tangential to the main aim of the experiments reported. They were taken to test if participants were aware of the greater efficacy of the gesture modality relative to the vocal modality. Our key dependent variable is communication success, and the extent to which communication success can be explained by modality and signal universality. So, because we have no specific hypothesis with regard to signal similarity and Interpreter confidence, we are reluctant to include this analysis in the manuscript, especially given the tight space constraints at Proc B, and the risk it may distract the reader from our key message. We again note that our data are available on the OSF, so those interested in exploring this aspect of the data can do so.

22. **R3** notes that we did not include skipped trial descriptive statistics by group.

This point is well-taken, and applies to each of the dependent variables (DV) reported. We have added a table of descriptive statistics for each DV to the supplementary materials.

23. **R3** queried how skipped trials might have affected our results.

Avoiding words that participants found difficult to communicate would improve communication success. However, this effect is likely to be negligible in Experiment 1, where skipped trials were rare, there was no difference in the rate of trial skipping between the groups and modalities, and communication success was substantially below ceiling. In Experiment 2 the words communicated were more complex, so, unsurprisingly, trial skipping increased in frequency, although communication success was still substantially lower than ceiling. Here, there was no difference in the rate of trial skipping between the groups, but trial skipping was higher in the vocal modality, consistent with communication being more challenging in this modality compared to the gesture modality. Despite skipping the more challenging vocal trials, communication success was higher in the gesture modality. We believe this strengthens support for the primacy of the gesture modality in the early stages of language creation.

24. **R3** queried why there was no effect of similarity on communication success for gesture (when collapsed across groups in Experiment 2) and the effect was larger for vocalizations.

Across Experiments 1 and 2, five of the six correlations returned a statistically significant positive correlation between signal universality and communication success. The non-significant relationship **R3** notes is moderate in size by contemporary guidelines in psychological science (Gignac & Szodorai, 2016), and in all likelihood does not reach statistical significance due to the small sample size and lack of statistical power. The stronger correlation in the vocal modality may arise due to the greater variation in communication success. Importantly, taken together the evidence indicates that signal universality is important to communication success.

Gignac GE, Szodorai ET. 2016 Effect size guidelines for individual differences researchers.

Personality and Individual Differences **102**, 74–78. (doi:10.1016/j.paid.2016.06.069).

Discussion

25. As summarised by the Editor, **R3** noted that the discussion is fairly speculative and does not adequately circle back to the language origin question the paper set out to test.

This point is well-taken. The discussion has been thoroughly revised with this in mind (see points below for specific information). Importantly, we have added a conclusion section that returns to the language origin question the paper sets out to test.

26. **R1** requested that we discuss in more detail why we should rule out a multimodal-first approach to language.

It is not our intention to rule out a multimodal-first account of language origin. We have now made this clear in the discussion, and suggest future research that can test a multimodal-first account (lines 438-449).

27. **R1** suggested we rephrase ‘...communicate meanings across minds’ (page 10) which they interpreted as telepathy.

We have rephrased to ‘...communicate meanings across people’ (line 382).

28. **R2** suggested we move our discussion of embodied theories of cognition from the discussion to the introduction to help motivate our predictions around signal universality.

This is an invaluable suggestion. Its implementation provides a strong rationale for Experiment 2 and foreshadows our discussion of embodiment in the discussion section. This change lessened the speculative aspect of the discussion that **R3** raised (point 25).

29. **R2** indicated that we could have done more in the discussion section to explain how the differences between the two cultures (Australia and Vanuatu) might have affected the results.

The broader coverage of the literature on signal salience and embodied theories of cognition in the introduction and discussion helped address this concern. In addition, we have added an example of a gestured signal that is shared across cultures (for the word 'lock') to the original unshared gestured signal example (for the word 'chain'). This is used to highlight how environmental similarities and differences between the Australian and Ni-Vanuatu group might have affected the signals produced and their success (lines 412-420).

30. **R2** appreciated the videos we made available on the OSF, but noted there were only examples for nouns in the gesture modality. **R2** requested we add examples for the other word types (verbs and adjectives) and for the signals produced in the vocal modality

This is a valuable suggestion. We have now uploaded 12 examples of gestured signals and 12 examples of vocal signals to the OSF (<https://osf.io/36jpy/>). For each modality we provide 4 noun examples, 4 verb examples and 4 adjective examples (2 from Australian Producers and 2 from Ni-Vanuatu Producers that are matched on the word being communicated). Including more examples further illustrates the cultural similarities and differences **R2** noted in point 29.

31. **R3** notes that, "The embodied cognition explanation of the effects is a stretch in a variety of ways. It's not clear how it relates to specific theories of embodiment. It also doesn't seem to have any clear connection to the data in the experiment or data from prior work."

The revised introduction makes this connection clear.

32. Related to point 25 and 31, **R3** states, "Likewise, the claim that people in the same culture use similar gesture[s] because they live in a more similar environment [is] highly speculative. There are many other alternative explanations. For example, people living in the same culture learn conventional gestures of that culture."

The revised introduction and the results from the severely vision-impaired group address this issue. Without a shared visual experience (or a greatly diminished visual experience), convention is an unlikely explanation for the similarity of the gestured signals used by vision-impaired and sighted participants, and are instead consistent with an embodied explanation. Furthermore, a convention account would predict that communication success would be higher for the signals produced by sighted Australian Producers given that they were interpreted by sighted Australian Interpreters from the same culture. Added to this, the expanded cross-cultural examples support the influence of the environment on the expression of embodied signals.

33. **R3** suggests the issue of iconicity should be raised in the introduction. They also query whether 'universality' and 'iconicity' might be the same thing.

We now raise the issue of iconicity in the introduction and point to recent research showing that iconicity and universality are related constructs but are not the same thing (lines 73-85).

34. In response to the sentence, "In the vocal modality the opportunity for embodiment is absent or greatly diminished", **R3** noted that it is not clear how sounds can be embodied.

This sentence has been revised to, "In the vocal modality the opportunity for embodiment is absent".

We thank the reviewers for their time and very valuable feedback on our manuscript - we believe the manuscript is much improved following the revisions based on their feedback. We hope the changes to the manuscript are sufficient to warrant publication in *Proceedings B*.

Best regards

Appendix C

Dear Editor (and authors),

I have re-read the manuscript with great interest and I find that it has improved significantly and has made it even stronger than the first submission. The authors have taken on board all the suggestions and they have addressed them in great level of detail. The changes that have been made regarding my comments are explained below:

1. Naturalness and universality: The authors have followed the reviewers' suggestion and have removed the word 'naturalness' from the title. They opted for a more nuanced title that really captures the essence of the key findings and removes grandiose claims.
2. Multimodality. The authors have included a paragraph at the end of the discussion (lines 663-674) where they explicitly state that they do not rule out a multimodal-first account for language emergence. They highlight the contradicting evidence and they suggest future lines of research to further our understanding of the origins of language as we know it today.
3. Blind participants. The authors include a very strong justification for the inclusion of blind participants in their study (lines 105-115).
4. Multiple choice task. The authors have included a good justification for the use of this task. One could argue that if they used an open-cloze task they could use a similar scoring method to objectively measure accuracy in responses as they did for the similarity across gestures (i.e., the researchers could score on a scale from 1-6 how similar are the guesses made for each gesture/vocalisation). Granted, it would be time-consuming but it could be done. I just want to highlight that by using the multiple choice format the authors are already constraining the options for the Interpreters and if the options are not well selected participants could perform at ceiling level. For instance, if the options for the gesture 'to drink' were *cat, house, freedom, drink* chances are that participants won't get it wrong. Perhaps the authors may want to add a line or two explaining how the options in the multiple choice task were selected? The editor may decide if this is required.
5. Gestures and signs. The authors have elaborated on the distinction between gesture and sign.
6. I was particularly enthusiastic about the following argument: 'To the extent that people physically interact in similar environments, it follows that they will manually represent their environment in similar ways, giving rise to representations that are shared across people'. This proposal is in line with our own work on silent gesture where we find that people from different cultures (i.e., Dutch and Mexican) come up with strikingly similar gestures due to our embodied knowledge of the world (Ortega & Özyürek, 2020). For the same reason, we also see that many sign languages of the world have very similar forms to express concepts even when the languages are not related (e.g., the sign TO-DRINK is very similar in many sign languages). It is exciting to see that the current manuscript provides more evidence that our body and

how we interact with the world may have been (the most?) important semiotic tool to give rise to language.

Ortega, G., & Özyürek, A. (2020). Types of iconicity and combinatorial strategies distinguish semantic categories in silent gesture across cultures. *Language and Cognition*, 12(1). <https://doi.org/10.1017/langcog.2019.28>

One outstanding issue may be to check the availability of the vocalisations/gestures in OSF. I tried to see some of the videos but I couldn't see anything. Maybe the setting of the website are on private?

I would like to reiterate that this is an impeccably designed study, stringent statistical analysis and with strong theoretical motivation. I am satisfied with the corrections and would like to endorse it for publication. I am sure this paper will make an important contribution to field and will be appealing to a broad readership.

Dr. Gerardo Ortega
University of Birmingham

Appendix D

THE UNIVERSITY OF
WESTERN AUSTRALIA
Achieving International Excellence

School of Psychological Science
35 Stirling Highway, CRAWLEY
Perth, Western Australia, WA 6009
T +61 (0)8 6488 2688
F +61 (0)8 6488 1006
E nicolas.fay@gmail.com
CRICOS Provider Code: 00126G

14 February 2022

The Editor, *Proceeding of the Royal Society B*

Dear Professor Barton,

Thank you for once again overseeing the review of our manuscript. We are delighted that our manuscript has now been accepted for publication at *Proceedings B*. Each of the minor revisions suggested have been implemented.

Minor Revisions

1. As suggested by the editor, the last line of the **abstract** has been changed to, “Our results support the hypothesis that gesture is the primary modality for language creation.” (line 39 of the **tracked changes document**).
2. As suggested by the editor, the last line in the **conclusion** section has been edited to, “The universality of gesture means it is ideally suited to bootstrapping human communication among modern humans, and therefore supports the hypothesis that gesture is the primary modality for language creation.” (lines 469-470).
3. As suggested by the editor, the word “ambitious” has been removed (line 49).
4. The editor noted that wording incorrectly suggests that it is not possible to compare communication across modalities in general in nonhumans. This sentence has been changed to, “However, because meaning is operationalised differently in primate vocal and gesture studies, the findings cannot currently be compared across the modalities.” (line 62).
5. As suggested by the editor, the sentence has been edited to, “The experiments reported here aim to overcome these issues.” (line 76)
6. Reviewer 1 (Gerardo Ortega) requested suggested we acknowledge that communication success may be inflated due the MCQ format used (the selection procedure for the MCQ options is detailed in the methods sections). We have added a sentence to this effect, “We note that communication success may be inflated in the multiple-choice format used, relative to a more open-ended format (although the inflation should be the same for both the gesture and vocalisation trials).” (lines 203-205).
7. Reviewer 1 (Gerardo Ortega) noted he could not access the example gestures/vocalisations made available on the OSF. I mistakenly left access as private. It is now public.
8. Additional minor edits have been made to improve readability, plus an Acknowledgements section and a Data Availability section have been added.

We thank the Editor and Gerardo Ortega for their time and valuable feedback on our manuscript, and look forward to seeing our paper in *Proceedings B*.

Best regards